# Cytogenetic Mapping of 35 New Markers in the Alpaca (*Vicugna pacos*)

**DOI:** 10.3390/genes11050522

**Published:** 2020-05-08

**Authors:** Mayra N. Mendoza, Terje Raudsepp, Manuel J. More, Gustavo A. Gutiérrez, F. Abel Ponce de León

**Affiliations:** 1Instituto de Investigación en Bioquímica y Biología Molecular, Universidad Nacional Agraria La Molina, Lima, Peru; Mayra.Mendoza.Cerna@outlook.com (M.N.M.); mmoremontoya@gmail.com (M.J.M.); 2Molecular Cytogenetics and Genomics Laboratory, Texas A&M University, College Station, TX 77845-4458, USA; 3Animal Science, University of Minnesota, St. Paul, MN 55108, USA; apl@umn.edu

**Keywords:** FISH, cytogenetic map, fiber genes, SNPs

## Abstract

Alpaca is a camelid species of broad economic, biological and biomedical interest, and an essential part of the cultural and historical heritage of Peru. Recently, efforts have been made to improve knowledge of the alpaca genome, and its genetics and cytogenetics, to develop molecular tools for selection and breeding. Here, we report cytogenetic mapping of 35 new markers to 19 alpaca autosomes and the X chromosome. Twenty-eight markers represent alpaca SNPs, of which 17 are located inside or near protein-coding genes, two are in ncRNA genes and nine are intergenic. The remaining seven markers correspond to candidate genes for fiber characteristics (*BMP4, COL1A2, GLI1, SFRP4)*, coat color (*TYR*) and development (*CHD7, PAX7*). The results take the tally of cytogenetically mapped markers in alpaca to 281, covering all 36 autosomes and the sex chromosomes. The new map assignments overall agree with human–camelid conserved synteny data, except for mapping *BMP4* to VPA3, suggesting a hitherto unknown homology with HSA14. The findings validate, refine and correct the current alpaca assembly *VicPac3.1* by anchoring unassigned sequence scaffolds, and ordering and orienting assigned scaffolds. The study contributes to the improvement in the alpaca reference genome and advances camelid molecular cytogenetics.

## 1. Introduction

The alpaca (*Vicugna pacos*, VPA) is a South American camelid adapted to the Andean highlands and domesticated by native people about 6000–7000 years ago [1,2,3]. Over 85% of the world alpaca population lives in Peru, where the species is a symbol of cultural heritage but also of high economic importance [4,5,6]. Alpaca fiber, which is valued for its softness and resistance, is an important export item for Peru, and has made alpacas a popular livestock species worldwide [7]. Besides, alpacas are valued as docile companion species and potential therapy animals [8,9]. Together with other camelids, alpacas are also of particular biological, biomedical and evolutionary interest due to their adaptations to extreme environments [10,11], unique and unusual features of their adaptive immune system [12,13,14,15] and as a basal clade of Cetartiodactyla in the mammalian phylogenetic tree [16,17]. Furthermore, the evolutionary history, genetic relationships and population structure of the alpaca and other South American camelids (llama, *Lama glama*; guanaco, *Lama guanicoe* and vicuña, *Vicugna vicugna*), continue to be topics of interest and debate [18,19,20]. 

Despite being a species of cultural, economic and scientific importance, the progress of alpaca genomics and the development of molecular tools for selection and breeding has been delayed compared to other livestock species. That is why the first systematic attempts to identify single nucleotide polymorphisms (SNPs) in the alpaca genome took advantage of the bovine high-density SNP array (BovineHD Genotyping BeadChip; Illumina: https://www.illumina.com/Documents/products/datasheets/datasheet_bovineHD.pdf). In the first study, SNPs were identified by genotyping alpaca radiation hybrid panel clones on the bovine array [21] and, in the second, by genotyping 40 individual alpacas [4]. The latter work identified 6756 alpaca SNPs, of which 400 were unique and polymorphic and 209 were located in genes. However, only 292 SNPs were assigned to alpaca chromosomes based on the dromedary-cattle–human conserved synteny data [22] and the alpaca whole genome cytogenetic map [23]. The first alpaca chromosome-level reference genome *VicPac3.1* became available only recently [11], shortly after these SNP discoveries. While *VicPac3.1* essentially improves the previous assemblies and assigns 76% of the genome to the 36 pairs of alpaca autosomes and the X chromosome, 24% of the genome still remains unplaced. Furthermore, the chromosomally assigned scaffolds remain unlocalized within a chromosome, thus providing only limited information about the relative order and orientation, and the exact cytogenetic location of specific genes and markers. 

Fluorescence in situ hybridization (FISH) is a well-established approach for determining the location and relative order of DNA sequences in chromosomes [24,25]. Cytogenetic maps remain useful in the genome sequencing era by anchoring sequence scaffolds to chromosomes, and for refining and validating sequence assemblies [11,23]. This is particularly important in species like camelids where the very recently emerged chromosome-level assemblies for the alpaca [11] and the dromedary [26] had only limited support from cytogenetic data. The current alpaca/camelid cytogenetic map comprises less than 250 markers [5,23,27,28,29], which is low considering the high diploid number (2n = 74) and when compared to the cytogenetic maps of other domestic species. 

The aims of this study are to improve the alpaca cytogenetic map and the recent genome assembly *VicPac3.1* by FISH mapping 35 new markers, of which 28 represent alpaca SNPs and seven correspond to genes associated with fiber characteristics, coat color and development. 

## 2. Materials and Methods

### 2.1. Ethics Statement

Alpaca blood samples for cell cultures and chromosome preparations were obtained in accordance with the United States Government Principles for the Utilization and Care of Vertebrate Animals Used in Testing, Research and Training. These protocols were approved as AUP #2018-0342 CA at Texas A&M University.

### 2.2. Selection of Markers

A recent study identified 400 unique polymorphic SNPs in alpacas using BovineHD Genotyping Beadchip [4]. Of these, 209 SNPs were not chromosomally assigned based on the comparative [22] and cytogenetic map [23] information. Flanking sequences of the unassigned SNPs were aligned to *VicPac3.1* [11], and only 141 SNPs were found in chromosomally assigned scaffolds. Of these, 101 SNPs were chosen for final analysis (Appendix A). Since several SNPs co-localized in the same scaffold, only one SNP was selected for cytogenetic mapping per smaller (<12 Mb) scaffolds. For larger scaffolds (from 12 Mb to 120 Mb), if possible, two or three SNPs located at the 5’ and 3’ ends, and in the center, were selected. As a result, out of the initial 101 SNPs, 28 SNPs were selected for FISH mapping (Table 1 and Appendix A).

Flanking sequences for each of the 101 SNPs were re-aligned by BLAST (NCBI: https://blast.ncbi.nlm.nih.gov/Blast.cgi) to update their location with respect to nearby genes. For SNPs that were not localized within gene sequences, information about the most proximal gene was retrieved. Finally, flanking sequences of all 101 SNPs were aligned for a second time by BLAST with *VicPac3.1* to confirm their chromosomal assignment and sequence position within scaffolds. For this, *VicPac3.1* was converted into a BLAST database using BLASTplus (NCBI: https://www.ncbi.nlm.nih.gov/books/NBK279668/), and sequences were aligned locally with Megablast. Composite information about all markers used in this study is presented in Appendix A.

We also selected, for cytogenetic mapping, seven candidate genes for alpaca traits of interest (Table 1, Appendix A). Information about these genes was retrieved from publications: *BMP4, COL1A2, GLI1, SFRP4* as candidate genes for fiber growth characteristics [30], *TYR* as a dilution gene for fiber color [31], *PAX7* as a regulator of neural crest development [32], and *CHD7* as a putative candidate gene for choanal atresia in alpacas [33]. 

Altogether, 35 markers were selected for BAC library screening and FISH mapping. These included 28 SNP-based markers and seven genes.

### 2.3. Design of PCR Primers and Overgo Probes

Sequences flanking the selected SNPs and sequences specific for the seven genes were retrieved from the NCBI Genome (https://www.ncbi.nlm.nih.gov/genome). Primers for PCR were designed with Primer3 [34] and Primer-BLAST (https://www.ncbi.nlm.nih.gov/tools/primer-blast/). All primers were validated by in silico PCR in UCSC Genome Browser (https://genome.ucsc.edu/) and optimized on alpaca genomic DNA. Overgo probes were designed manually, as described previously [5]. PCR and overgo primers for each marker are presented in Appendix A. 

### 2.4. Screening Alpaca CHORI-246 BAC Library and BAC DNA Isolation

Radioactively, [^32^P] labeled overgo primers were hybridized to the high-density filters of CHORI-246 alpaca genomic BAC library (https://bacpacresources.org/). The filters were exposed to autoradiography and positive BAC clones were identified and picked from the library, as described elsewhere [5,35]. BACs corresponding to individual markers were identified by PCR with marker-specific primers (Appendix A). BAC DNA was isolated with the Plasmid Midi Kit (Qiagen, Germantown Road Germantown, MD, USA) following the manufacturer’s protocol and evaluated for quality by electrophoresis in 1% agarose gels.

### 2.5. Cell Cultures and Chromosome Preparations

Metaphase and interphase chromosome preparations were made from short-term peripheral blood lymphocyte or primary fibroblast cell cultures following standard protocols [23,25,35]. Alpaca blood lymphocytes were stimulated into proliferation with concanavalin A, a mitogen from *Canavalia ensiformis* (20 μg/mL; Sigma Aldrich, St. Louis, MO, USA). Cells were harvested with demecolcine solution (0.1 μg/mL; Sigma Aldrich), treated with optimal hypotonic solution (Rainbow Scientific, Maple Avenue, Windsor, CT, USA), and fixed in 3:1 methanol/acetic acid. Approximately 10 µL of fixed cell suspension was dropped on precleaned wet glass slides at room temperature and air dried. The quality and quantity of metaphase spreads was evaluated under phase contrast microscope.

### 2.6. Fluorescence In Situ Hybridization (FISH) and Analysis

The DNA of individual BACs was labeled with biotin or digoxigenin using Biotin or DIG Nick Translation Mix (Roche Diagnostics), respectively, and the manufacturer’s protocol. In situ hybridization and signal detection was done following standard protocols described elsewhere [25,35]. Biotin-labeled probes were detected with avidin-FITC (Vector Laboratories) and dig-labeled probes with anti-DIG-rhodamine (Roche Applied Science). In order to precisely determine the cytogenetic location of the 35 new markers, each marker was co-hybridized with a differently labeled previously FISH-mapped reference marker [5,23] (Table 1 and Appendix A). Images for a minimum of 10 metaphase spreads and 10 interphase nuclei were captured for each experiment and analyzed using a Zeiss Axioplan 2 fluorescence microscope, equipped with the Isis Version 5.2 (MetaSystems GmbH, Altlussheim, Germany) software. Chomosomes were counterstained with 4′,6-diamidino-2-phenylindole (DAPI) and identified according to the previously proposed nomenclature [22,23,35].

## 3. Results

### 3.1. Bioinformatic Analysis of SNP Markers

Bioinformatic analysis by BLAST (https://blast.ncbi.nlm.nih.gov/Blast.cgi) of the flanking sequences of the 101 selected SNPs (28 for FISH mapping and 73 supporting SNPs) confirmed their location within or near known genes or in intergenic regions, but also refined annotations for 10 markers (Appendix A). Among the SNPs selected for mapping, annotation was improved for five markers: intergenic UNA_272, UNA_369 and UNA_396 corresponded now to putative noncoding RNA (ncRNA) sequences; UNA_353 was located in the *AGBL1* gene instead of LOC107032903 [4], and the nearest gene, LOC102535064, to UNA_077 was annotated as neuroligin-1 (*NLGN1*) (Table 1, Appendix A). Altogether, among the 28 SNP markers selected for cytogenetic mapping, 17 were located inside or near protein coding genes, two were in ncRNA genes and nine were intergenic (Table 1).

### 3.2. Identification of Alpaca BACs Containing Specific SNPs and Genes

Altogether, we identified 121 BAC clones that collectively contained the 35 markers (28 SNPs and seven genes) of interest. The number of clones per marker ranged from one to seven, with only one BAC found for UNA_396, UNA_441, *CHD7* and *COL1A2*, and seven BACs found for UNA_114 and UNA_211 (Appendix A). One BAC clone per each marker was used for FISH mapping (Table 1), with the exception of *BMP4,* where both BACs found for this gene were used.

### 3.3. Cytogenetic Mapping and Improvement of the Genome Assembly

All 35 genes and markers were assigned to specific bands and regions distributed in 19 alpaca autosomes and the X chromosome (Table 1). Co-hybridization of each new marker with a previously mapped reference marker confirmed chromosomal assignment and helped to position new markers in the centromere-telomere field (Figure 1, Appendix A). 

The majority of markers mapped to the chromosomes and chromosomal regions are in agreement with human–dromedary Zoo–FISH [22] and the assignment of scaffolds in *VicPac3.1* [11]. In the case of VPA1, FISH mapping essentially expanded conserved synteny block with HSA21 and, respectively, reduced conserved synteny region with HSA3 (Figure 1A). This is because the newly mapped gene, *CXADR,* from a large scaffold 4 (>27 Mb) and previously mapped *DYRK1A* [23] both map to HSA21 in humans, thus expanding homology segment with HSA21 to VPAq24-q33. The only true discrepancy, however, was mapping the *BMP4* gene from HSA14 to VPA3 (Figure 1B, Table 1) instead of VPA6, which shares a conserved synteny with HSA14 [22]. The location of *BMP4* in VPA3q13 was confirmed by mapping two BACs found for this gene (Table 1) in combination with three reference markers from VPA3 and two from VPA6 (Figure 1B; Appendix A). 

Altogether, the results confirmed the chromosomal location of 26 *VicPac3.1* scaffolds (Table 1), but in several cases also refined it and provided novel information. In six chromosomes, VPA1, 4, 11, 12, 26 and X, the newly mapped markers represented two different scaffolds and, thus, resolved their relative order along the chromosome (Figure 1A,C,E and Appendix A). Furthermore, FISH mapping UNA_441 to VPA26 resolved the previous ambiguous assignment of scaffold 4140 to either VPA9 or VPA26 (Figure 1E). In the case of VPA12, mapping *GLI1* anchored a previously unassigned scaffold 77319 to this chromosome and, together with mapping UNA_110, ordered scaffolds 77319 and 77306 along VPA12 (Figure 1C). 

Finally, the *VicPac3.1* chromosome and scaffold information for the 35 FISH mapped markers was analyzed together with an additional 73 SNP markers from these chromosomes (19 autosomes and the X). This confirmed the chromosomal/scaffold assignment of all 108 markers and genes, but also determined the telomere–centromere orientation of 22 scaffolds in 19 alpaca autosomes (Figure 1, Appendix A).

Collectively, the results enriched the alpaca cytogenetic map, and validated and refined the current genome assembly. The mapping of *BMP4* suggests the presence of a hitherto unknown conserved synteny segment between VPA3 and HSA14.

## 4. Discussion 

We report on the cytogenetic mapping of 35 new markers in the alpaca genome. The results improve FISH maps of 19 autosomes and the X chromosome with gene-specific markers, and for the first time, with markers corresponding to polymorphic alpaca SNPs. In addition, chromosomal mapping of these 28 SNPs confirmed and refined the assignment and locations of another 73 polymorphic SNPs from the same scaffolds or chromosomes (Appendix A). Altogether, these 101 SNPs have been reported as shared markers between alpaca and cattle [4], and are expected to be conserved in all South American camelids, and maybe even in Old World camels. Knowledge about their genomic distribution and precise chromosomal location will assist the systematic selection of SNPs for designing whole-genome genotyping platforms for the alpaca and related species. 

Cytogenetic mapping in the alpaca has been moderate compared to other domestic species. Partially, this is because of the high diploid number (2n = 74) and difficulties in unambiguously identifying chromosomes [35]. Therefore, confident FISH mapping of new markers requires their co-hybridization with validated references, as shown in this and previous studies [5,23,27,35]. The main source of FISH probes, the alpaca genomic BAC library CHORI-246 (https://bacpacresources.org/), has not been pooled for screening by PCR. Therefore, the identification of BACs of interest is done by hybridization of the library filters with isotope-labeled oligos, which is tedious and requires certified laboratory settings. Likewise, the alpaca BAC library has not been a subject for BAC end sequencing (BES), due to which the BACs cannot be aligned with the reference assembly to facilitate finding clones of interest. In contrast, such tracks of overlapping BAC clones are available for other domestic species (NCBI genome: https://www.ncbi.nlm.nih.gov/genome/) and have been successfully used, for example in horses, for anchoring unassigned scaffolds by FISH [36,37] and resolving complex genomic regions [38]. Nevertheless, the alpaca cytogenetic mapping, which started more than a decade ago by the assignment of cosmid clones for immunoglobulin heavy chain (*IGH*) locus [29], has gradually developed into a whole-genome (WG) map covering all autosomes and the sex chromosomes [23,35]. Since then, the initial WG map with 230 markers [23] has made moderate progress, now comprising 281 specific genes, SNPs and sequence tagged sites (STSs) (Figure 2). Of the 51 newly added markers, 16 were published last year [5,27,28] and 35 reported in this study. Notably, the recent mapping of four casein genes to camelid chromosome 2 (VPA2, Figure 2) successfully used a pool of PCR amplicons over the gene cluster as a FISH probe [28], providing a viable alternative to BACs for certain cases.

Since the beginning, an important backbone for the alpaca cytogenetic map has been the conserved synteny (Zoo-FISH) information with humans [22], which has guided the systematic selection of markers for mapping, as well as validating the results in this and previous studies [5,23,27,35,39]. The locations of the majority of FISH-mapped markers have been in good accordance with Zoo-FISH, but also refine it by providing information about gene order within conserved synteny segments and more accurately demarcating segment boundaries [23]. An example from this study is the refinement of the boundaries of conserved synteny with HSA3 and HSA21 in VPA1 (Figure 1A). However, in a few cases, FISH mapping has provided Zoo-FISH with missing data, like revealing conserved synteny between VPA36 and HSA7 [39], or discovering hitherto unknown conserved syntenies, like the recent mapping of *MC1R* from HSA16 to VPA21q (Figure 2) [27], the region previously thought to share synteny with HSA1 only [22]. This new conserved synteny of HSA1 and HSA16 in camelid chromosome 21 is also supported by the recent chromosome-level dromedary genome assembly *CamDro3* [26]. However, mapping of the *BMP4* gene from HSA14 to VPA3 in this study (Figure 1B) was more problematic. According to Zoo-FISH [22], *VicPa3.1* [11] and *CamDro3* [26], camelid chromosome 3 shares conserved synteny with HSA5 only, while conserved synteny with HSA14 is limited to camelid chromosome 6 (Figure 2). The co-hybridization of both *BMP4*-containing BACs (17I20 and 63L14; Table 1, Appendix A) with multiple VPA3 and VPA6 reference markers (Appendix A), clearly assigned *BMP4* to VPA3 (Figure 1B). Since *BMP4* sequence identity was well-confirmed by in silico PCR and BLAST analysis of the PCR product, we infer the presence of a small, hitherto unknown, conserved synteny segment of HSA14 in VPA3, intervening with the large region of synteny with HSA5 (Figure 1B). Indirectly, this is supported by the chimeric assignment of a *BMP4* sequence in *VicPac3.1* to two different scaffolds: a large scaffold 77258 in VPA6 with homology to HSA14, and an unassigned scaffold 33 [11]. It is possible that the latter is the true location of *BMP4* and a missing part of VPA3 assembly, as indicated by our FISH results. Nevertheless, the assignment of *BMP4* and segmental homology of HSA14 with VPA3 remain tentative and require additional evidence by sequencing the FISH-mapped *BMP4* BACs and further improving the contiguity of the alpaca genome assembly with an additional long-read sequence and Hi-C chromatin interaction data. 

These and other recent technological advances have essentially improved the quality of mammalian genome assemblies. Therefore, it is anticipated that the current, rather fragmented, *VicPac3.1* [11] will very soon be replaced by a more contiguous assembly, ideally comprising a single scaffold per chromosome. However, the need for cytogenetic anchors will not disappear completely. As experienced from well assembled human and domestic animal genomes, FISH remains a viable method for chromosomal anchoring of unassigned sequences, which typically represent complex, copy number variable (CNV), segmentally duplicated or ampliconic regions [36,37,40,41]. In specific cases, FISH has been successfully used to validate and refine bioinformatically detected evolutionary chromosome changes between closely related species, such as cattle and goat [40], or as a tool for identifying individuals heterozygous for clinically important, large copy number variations in horses [42]. 

Another important aspect of cytogenetic mapping is that it expands the collection of annotated BAC clones. Given that, on average, 2-4 BACs correspond to each FISH-mapped marker, the current 281-marker alpaca cytogenetic map (Figure 2) is accompanied by a genome-wide collection of more than 1000 BACs, of which 121 were identified in this study. This is a cost-effective resource for resolving the assembly of regions of complex genomic architecture, such as gene families, GC-rich regions, segmental duplications and CNVs. Instead of still quite expensive WG long-read sequencing, an accurate reconstruction of such regions can be obtained by a combined short- and long-read sequencing of a regional BAC tiling path. This approach has been successful for complex regions in human [43] and animal [38,42] genomes. The collection of alpaca BAC clones corresponding to specific genes, SNPs and STSs, as identified in this and previous studies [5,23,27], has a potential to serve the same purpose for camelid genomes.

Finally, the improved alpaca WG cytogenetic map and the expanded collection of annotated BAC clones serve as important tools for clinical cytogenetics for any camelid species. In contrast to most other domestic animals, chromosome identification by size, morphology and banding patterns has limited success in camelids and requires molecular tools [35,39,44]. An example from this study was mapping markers to VPA29 (Appendix A), where the homologs showed extensive differences in size and morphology and could only be identified by FISH. 

## 5. Conclusions and Future Approaches

The present study improved the alpaca whole genome cytogenetic map with 35 new markers, taking the total tally to 281 FISH-mapped markers in this species. Overall, the alpaca cytogenetic map stays in good agreement with the known human–camelid Zoo-FISH homology [22], though the assignment of *BMP4* to VPA3 in this study suggests a novel segment of conserved synteny and needs further investigation. The findings also added 121 BAC clones to the collection of approximately 1000 annotated alpaca BACs. We showed that the cytogenetic map continues to be an important resource for validating, refining and correcting the current alpaca sequence assembly *VicPac3.1*. The collection of BACs, on the other hand, is a potential tool for resolving misassembled regions and regions of complex genomic architecture. Following the trends in other species, it is anticipated that the application of the advanced genomics technologies will soon take the alpaca genome assembly to the next qualitative level. Despite this and, again, based on examples from other species, cytogenetic map and a collection of FISH probes remain viable tools for validating, correcting and fine-tuning the assembly. Furthermore, this is a two-way road because a good quality reference genome is the prerequisite for the design of next-generation, oligonucleotide-based FISH probes [45,46] to further advance both camelid genomics and clinical cytogenetics.

## Figures and Tables

**Figure 1 genes-11-00522-f001:**
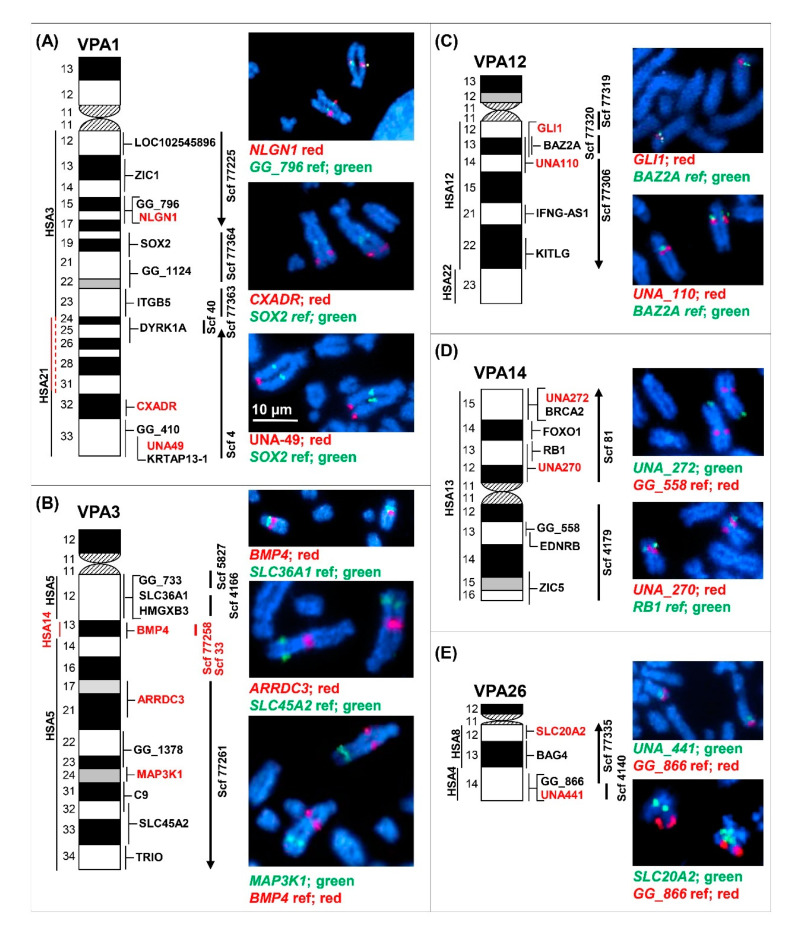
**Selected examples of fluorescence in situ hybridization (FISH) mapping in five alpaca chromosomes:** (**A**) VPA1; (**B**) VPA3; (**C**) VPA12; (**D**) VPA14, and (**E**) VPA26. Chromosome ideograms with cytogenetic nomenclature, conserved synteny with human and all cytogenetically mapped markers (new markers in red font) are shown on the left; vertical lines in the middle show corresponding *VicPac3.1* scaffolds; lines with arrowheads indicate orientation, and microscope images with FISH results are on the right.

**Figure 2 genes-11-00522-f002:**
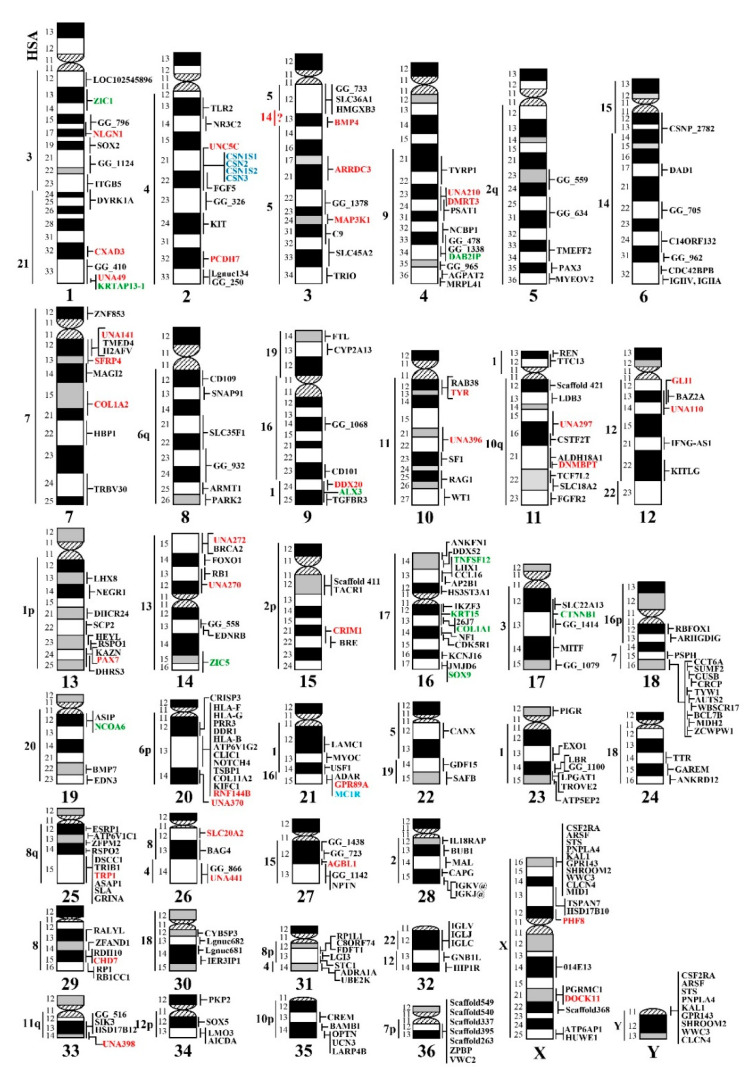
**Summary status of the alpaca/camelid whole genome cytogenetic map with 281 markers.** Markers in red font were mapped in this study; black font—original WG map by Avila et al. (2014a); green font—mapped by Mendoza et al. (2019); dark blue font—mapped by Pauciullo et al. (2019); light blue font—mapped by Alshanbari et al. (2019).

**Table 1 genes-11-00522-t001:** Summary information about the cytogenetically mapped SNP markers and genes; *—assignment tentative.

Cytogenetic Location	SNP ID More et al. 2019	Gene Symbol	Scaffold *VicPac3.1*	Chromosome*VicPac3.1*	BAC for FISH, CHORI-246
1q16-q17	UNA_077	*NLGN1 (LOC102535064)*	77225	1	250G20
1q32	UNA_071	*CXADR*	4	1	203G23
1q33ter	UNA_049	intergenic	4	1	27E21
2q21-q22	UNA_172	*UNC5C*	26	2	113I16
2q32	UNA_174	*PCDH7*	26	2	31H13
3q13	n/a	*BMP4* *	77258; 33	6	17I20, 63L14
3q17-q21	UNA_150	*ARRDC3*	77261	3	87L21
3q24	UNA_345	*MAP3K1*	77261	3	106K2
4q23-q24	UNA_211	*DMRT3*	8430	4	41L6
4q23-q24	UNA_210	intergenic	2524	4	46F18
7q12	UNA_141	intergenic	8475	7	83F11
7q12-q13	n/a	*SFRP4*	8475	7	190L6
7q15-q21	n/a	*COL1A2*	8475	7	43N9
9q24	UNA_116	*DDX20*	160	9	68O18
10q12-q13	n/a	*TYR*	127	10	74O16
10q21-q22	UNA_396	putative ncRNA	127	10	196K17
11q15-q16	UNA_297	intergenic	77342	11	174G24
11q21ter	UNA_114	*DNMBP*	77343	11	62A20
12q12-q13	n/a	*GLI1*	77319	unassigned	85P24
12q13-q14	UNA_110	intergenic	77306	12	29N15
13q24-q25	n/a	*PAX7*	77224	13	29J22
14p12-p13	UNA_270	intergenic	81	14	22F1
14p15	UNA_272	putative ncRNA	81	14	71E16
15q21-q22	UNA_252	*CRIM1*	46	15	135P3
20q13ter	UNA_369	*RNF144B/* putative ncRNA	77293	20	10N10
20q14	UNA_370	intergenic	77293	20	418M11
21q15	UNA_095	*GPR89A*	77374	21	218E10
25q15	UNA_287	*TRPS1*	77333	25	45D13
26q12	UNA_442	*SLC20A2*	77335	26	114K20
26q14	UNA_441	intergenic	4140	9,26	164M19
27q12ter	UNA_353	*AGBL1*	11	27	15N15
29q15	n/a	*CHD7*	3	29	431P24
33q13-q14	UNA_398	intergenic	12	33	13A11
Xp12	UNA_413	*PHF8*	232	X	120H23
Xq21	UNA_399	*DOCK11*	93	X	13C22

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
