# Peer review of "Cytogenetic Mapping of 35 New Markers in the Alpaca (*Vicugna pacos*)"

_genes, 2020, doi:10.3390/genes11050522_

Round 1

Reviewer 1 Report

The manuscript by Mendoza et al. brings new information about the alpaca genome both at the molecular as well as at the chromosome level. It also provided novel information about evolutionarily conserved syntenies.

It is well written, concise, but still informative. I recommend some minor changes, which could improve the readability of the paper in some of its parts.

Lines 76-87: This paragraph should be re-written to better explain the experimental design and should be harmonized with the corresponding part(s) of Results. The process of selecting markers and its results are presented in a rather confusing way. In different parts of the manuscript, including its title, it is not always immediately clear which group of markers described in M&M are actually concerned. Although it can eventually be deciphered by the reader, such amendments would contribute to the readability of the text.

Besides clear improvements of the alpaca genome and resolution of some previous unclear assignments of scaffolds, an unexplained discrepancy of homologies/syntenies was identified for the BMP4 gene (lines 240-254).  I understand and agree with discussion (lines 251-254), but I recommend to add a similar sentence to the section Conclusions and Future Approaches to point out this interesting result and a possible follow-up.

Technical remark: Authors contributions, Acknowledgments and Conflict of Interest statement are missing, at least in the version that I have got.

Author Response

Comment 1: Lines 76-87: This paragraph should be re-written to better explain the experimental design and should be harmonized with the corresponding part(s) of Results. The process of selecting markers and its results are presented in a rather confusing way. In different parts of the manuscript, including its title, it is not always immediately clear which group of markers described in M&M are actually concerned. Although it can eventually be deciphered by the reader, such amendments would contribute to the readability of the text.

Response 1: We have revised paragraph 2.2 Selection of Markers, which now reads as follows:

2.2 Selection of Markers

 A recent study identified 400 unique polymorphic SNPs in alpacas using BovineHD Genotyping Beadchip [4]. Of these, 209 SNPs were not chromosomally assigned based on the comparative [22] and cytogenetic map [23] information. Flanking sequences of the unassigned SNPs were aligned to VicPac3.1 [11], and only 141 SNPs were found in chromosomally assigned scaffolds. Of these, 101 SNPs were chosen for final analysis (Table S1). Since several SNPs co-localized in the same scaffold, only one SNP was selected for cytogenetic mapping per smaller (< 12 Mb) scaffolds. For larger scaffolds (from 12 Mb to 120 Mb), if possible, two or three SNPs located at the 5’ and 3’ ends, and in the center were selected. As a result, out of the initial 101 SNPs, 28 SNPs were selected for FISH mapping (Table 1 and Table S1).

Flanking sequences for each of the 101 SNPs were re-aligned by BLAST (NCBI: https://blast.ncbi.nlm.nih.gov/Blast.cgi) to update their location with respect to nearby genes. For SNPs that were not localized within gene sequences, information about the most proximal gene was retrieved. Finally, flanking sequences of all 101 SNPs were aligned for a second time by BLAST with VicPac3.1 to confirm their chromosomal assignment and sequence position within scaffolds. For this, VicPac3.1 was converted into a BLAST database using BLASTplus (NCBI: https://www.ncbi.nlm.nih.gov/books/NBK279668/), and sequences were aligned locally with Megablast. Composite information about all markers used in this study is presented in Table S1.

We also selected for cytogenetic mapping 7 candidate genes for alpaca traits of interest (Table 1, Table S1). Information about these genes was retrieved from publications: BMP4, COL1A2, GLI1, SFRP4 as candidate genes for fiber growth characteristics [30], TYR as a dilution gene for fiber color [31], PAX7 as a regulator of neural crest development [32], and CHD7 as a putative candidate gene for choanal atresia in alpacas [33].

Altogether, 35 markers were selected for BAC library screening and FISH mapping. These included 28 SNP-based markers and 7 genes.”

Comment 2: Besides clear improvements of the alpaca genome and resolution of some previous unclear assignments of scaffolds, an unexplained discrepancy of homologies/syntenies was identified for the BMP4 gene (lines 240-254).  I understand and agree with discussion (lines 251-254), but I recommend to add a similar sentence to the section Conclusions and Future Approaches to point out this interesting result and a possible follow-up.

Response 2: Thank you. We added a sentence to the last section:

“Overall, the alpaca cytogenetic map stays in good agreement with the known human-camelid Zoo-FISH homology [22], though the assignment of BMP4 to VPA3 in this study suggests a novel segment of conserved synteny and needs further investigation”

Comment 3: Technical remark: Authors contributions, Acknowledgments and Conflict of Interest statement are missing, at least in the version that I have got.

Response 3: Thank you. These statements are added to the end of the manuscript, though we assumed that these will be automatically generated by the submission pipeline.

Reviewer 2 Report

The aims of the authors were to improve the alpaca cytogenetic map and the genome 
assembly VicPac3.1 by FISH mapping . They report about cytogenetic mapping of 35 new markers in the alpaca genome, of which 28 represent alpaca SNPs and 7 
correspond to genes associated with fiber characteristics, coat color and development. 

The chromosomal  mapping of 28 SNPs confirmed and refined assignment and locations of another 73 polymorphic SNPs from the same scaffolds or chromosomes. 

Fish mapping resolve previous ambiguous assignment.

Knowledge about 
genomic distribution and precise chromosomal location of the 101 SNPs reported as shared markers between alpaca and cattle, will assist systematic selection of SNPs for 
designing whole-genome genotyping platforms for the alpaca and related species. 

The paper is well written and concise in its conclusions.

Materials and Methods are described in detail and methodically.

However there are the following moderate observations.

- Paragraph 3.1:

this paragraph should be rewritten by including part of the information in the "Selection of markers" paragraph of the M&M. That is, it should be indicated here how they started for the selection of the 35 markers for the FISH.

- Paragraph 3.2: lines 149-150

“Altogether, we identified 121 BAC clones that collectively contained the 35 markers (28 SNPs and 7 genes) of interest”.

Paragraph 3.3: lines 185-187

“Finally, VicPac3.1 chromosome and scaffold information for the 35 FISH mapped markers was analyzed together with an additional 73 SNP markers from these chromosomes (19 autosomes and the X)”. 

- Discussion: line 194

“We report about cytogenetic mapping of 35 new markers in the alpaca genome”. 

 - Conclusions and future Approaches: lines 284-285

“The present study improved the alpaca whole genome cytogenetic map with 35 new markers, 
taking total tally to 281 FISH mapped markers in this species”. 

The title is misleading and should be changed according to the content of the sentence “in quotes” unless the Authors refer to the fact that 17 among the 28 SNP markers selected for cytogenetic mapping were located inside or near protein coding genes (lines 145-146).

- Line 141: Are these 10 markers highlighted in Table S1?

- Line 171: CXADR instead of CXAD3?

- Line 264: large copy number variations (CNV) instead of CNV.

Author Response

Comment 1: - Paragraph 3.1: this paragraph should be rewritten by including part of the information in the “Selection of markers” paragraph of the M&M. That is, it should be indicated here how they started for the selection of the 35 markers for the FISH.

Response 1: As per the comments of another reviewer, we essentially revised Section 2.2. Selection of Markers, which hopefully clarifies, how the selection of markers started and how we ended up with mapping 35 markers. As requested by this reviewer, we also made minor revisions in Section 3.1 Bioinformatic Analysis of SNP Markers to keep it in line with the revisions made in Section 2.2

Comment 2:

- Paragraph 3.2: lines 149-150

“Altogether, we identified 121 BAC clones that collectively contained the 35 markers (28 SNPs and 7 genes) of interest”.

“Finally, VicPac3.1 chromosome and scaffold information for the 35 FISH mapped markers was analyzed together with an additional 73 SNP markers from these chromosomes (19 autosomes and the X)”.

- Discussion: line 194

“We report about cytogenetic mapping of 35 new markers in the alpaca genome”.

- Conclusions and future Approaches: lines 284-285

“The present study improved the alpaca whole genome cytogenetic map with 35 new markers, taking total tally to 281 FISH mapped markers in this species”.

The title is misleading and should be changed according to the content of the sentence “in quotes” unless the Authors refer to the fact that 17 among the 28 SNP markers selected for cytogenetic mapping were located inside or near protein coding genes (lines 145-146).

Response 2: We changed the title to “Cytogenetic mapping of 35 new markers in the alpaca (Vicugna pacos)”

Comment 3: - Line 141: Are these 10 markers highlighted in Table S1?

Response 3: These 10 markers are now denoted with a star (*) in red font.

 Comment 4: - Line 171: CXADR instead of CXAD3?

Response 4: Corrected.

Comment 5: - Line 264: large copy number variations (CNV) instead of CNV.

Response 5: Corrected.